# An Overview of Sensors for Long Range Missile Defense

**DOI:** 10.3390/s22249871

**Published:** 2022-12-15

**Authors:** Simone Fontana, Federica Di Lauro

**Affiliations:** 1School of Law, Università degli Studi di Milano-Bicocca, 20126 Milano, Italy; 2Department of Computer Science, Università degli Studi di Milano-Bicocca, 20126 Milano, Italy

**Keywords:** missile defense, hypersonic missile, radar, infrared, satellite

## Abstract

Given the increasing tensions between world powers, missile defense is a topic that is more relevant than ever. However, information on the subject is often fragmented, confusing and untrustworthy. On the other hand, we believe that an informed overview of the current status is important for decision makers and citizens alike. A missile is essentially a guided rocket and therefore the term can be used to describe a very wide range of weapon systems. In this paper, we focus on long-range and intercontinental threats, which we believe are more important and problematic to defend against. We provide an overview of the two most common types of sensors, space-based infrared sensors and radars, and highlight their peculiarities and, most importantly, their drawbacks that severely limit their effectiveness.

## 1. Introduction

Missile defense is a topic that is more relevant than ever before. Accurate, detailed and reliable information on this subject is essential for both decision makers and citizens. However, high-quality information is usually unavailable or fragmented into many different sources that are not easily consulted. Academic papers produced by independent, i.e., nonmilitary, research institutions are scarce, with few exceptions. However, we believe that independent research on this topic is essential given the relevance and actuality of the issue.

This paper is intended to provide a reliable overview of sensor technologies for long-range missile defense. By “long range”, we refer to weapon systems with a range greater than 3000 km. A missile is essentially a guided rocket, i.e., a device powered by rocket engines. This category includes a very wide variety of weapons, from shoulder-fired missiles to intercontinental nuclear ballistic missiles. These have very different characteristics and therefore the corresponding defense systems are also very different. For this reason, we think that a division into categories is necessary for a scientific research activity. We decided to focus on long-range weapons for two reasons. First, there is evidence that current long-range missile defense systems may suffer from severe limitations and are much less effective than generally recognized [1,2,3]. Furthermore, given the different phases of flight of a long-range missile and the very large distance it travels, the cooperation of several types of sensor systems is required. Ultimately, we believe that this type of weapon is indeed the most important.

We analyze three different types of long-range threats. Ballistic missiles are the oldest, but still very current. They are able to reach very long distances thanks to a parabolic trajectory that passes most of the time outside the atmosphere and therefore without air resistance. Cruise missiles, on the other hand, fly through the atmosphere, usually at lower speeds and over shorter distances. However, their trajectory is highly unpredictable and therefore poses a major challenge to defense systems. Finally, hypersonic glide vehicles, often mistakenly referred to as hypersonic missiles, are a very recent threat that combines high maneuverability with long range and high speed.

Each of these three types of threats has its own peculiarities, but also common characteristics with the others, so that most defense techniques are applicable to all categories.

Two main types of sensors are used for missile defense: either radars or imaging infrared sensors on satellites. The former are highly accurate and may have high resolution. However, they have one major drawback: they cannot see beyond the horizon. This is very important for defense against long-range missiles, since their high speed up to Mach 20 [4] leaves very little time to intervene after being seen by radars. For this reason, the latter type of sensor is essential. Infrared sensors on satellites are capable of detecting missiles shortly after launch and can therefore provide very useful information for interception systems. However, radars are still necessary because infrared sensors do not yet have the necessary accuracy to control an interceptor system on their own.

With this paper, we would like to give an overview of current sensors for long-range missile defense. We hope that this type of work can help disseminate reliable knowledge on this topic and promote independent research activity in this important area. First, we introduce the three types of threats and describe their strengths, weaknesses and characteristics. This is followed by a brief introduction to missile defense. Since the paper focuses on sensors and not on defense systems as a whole, the latter are not described. However, a brief introduction to the general operation of such systems is necessary to understand the overview. The last two sections describe the operation of radars and space-based infrared imaging sensors in the context of missile defense, highlighting their specificities and, most importantly, their limitations, which require further research.

## 2. Long Range Missile Threats

The characteristics of a defense sensor system are closely related to the threats it must deal with. For this reason, in this section we provide an overview of current long-range missile threats. A missile is essentially a guided rocket, which is a vehicle that uses rocket engines for propulsion. A rocket engine generates thrust by ejecting gasses produced by combustion at high velocity. Rocket engines differ from jet engines, which are nevertheless used in some types of missiles, because the latter use oxygen as oxidant for combustion, while the former use other oxidants that are carried in the tank with the fuel. It should be noted that missiles do not necessarily have to be propelled for the entire duration of flight; moreover, they are often guided only during part of the flight. When a missile is propelled or guided is an important feature that distinguishes the different types of weapon systems.

Traditionally, ballistic missiles are classified by range (see Table 1). By ”long range” we mean threats with a range greater than 3000 km, which can fall into the IRBM and ICBM category of Table 1. We decided to focus on threats with this range because they have some peculiarities. First, their speed and trajectory make them the most difficult to defend against. While there are defenses such as the Patriot or THAAD systems that are at least partially effective against short- and medium-range threats [5,6,7], defending against long-range weapon systems still seems to be a very problematic and open issue [1,8]. In addition, short- and long-range weapon systems, even if they belong to the same weapon category, often have different characteristics.

### 2.1. Ballistic Missile

Ballistic missiles are so called because they follow a ballistic trajectory, i.e., an approximately parabolic trajectory. This is not the shortest path to a target and thus may seem less efficient than a straight path or a path that is curved and parallel to the ground. However, the reason for this choice is that it corresponds to the path of least energy and therefore maximizes the range of the missile. This is because, thanks to this trajectory, the missile travels most of the time outside the atmosphere and thus without air resistance. This is fundamental for achieving very long ranges, since a missile without air resistance dissipates very little energy and can therefore fly farther.

The trajectory of a ballistic missile can be divided into three phases, shown in Figure 1: boost phase, midcourse phase and re-entry or terminal phase [4,9,11,12,13].

The boost phase is the first and only phase in which the missile is propelled, usually by one or more rocket engines. Its objective is to bring the missile to the boundary or outside the atmosphere. During or at the end of this phase, the engines and fuel tanks separate from the rest of the missile, as they serve no further purpose and would represent dead weight for the rest of the flight. In addition, for simple ballistic missiles, the boost phase is the only phase during which the missile is guided by a navigation system [4]. Table 2 lists the burning time and burn-out altitude, i.e., the maximum time that rocket boosters are used and the corresponding altitude and the range for a few ballistic missiles. It should be noted that even for intercontinental ballistic missiles, such as the SS-18 Satan, the burning time is only 5 min.

An important feature of this phase is that the rocket engines generate a large amount of heat. This is critical for missile defense sensors because heat is an important indicator used by many different systems [14,15,16,17].

**Table 2 sensors-22-09871-t002:** Burning time ($t_{b}$), burn-out altitude ($h_{b}$) and max range (*R*) of various kind of missiles (from short to intercontinental range). Data from [9,11,18]. The max range depends on the payload. Since some missiles have been developed to carry various kinds and numbers of warheads, the stated max range is that corresponding to the minimum payload.

Missile	$t_{b}$ [s]	$h_{b}$ [km]	*R* [km]
SS-1 Scud-B	70	30	300
DF-3A	140	100	3000
LGM-118 Peacekeeper	180	200	9600
MGM-134 Midgetman	220	340	11,000
Minuteman III	187	190	14,000
SS-18 Satan	300	400	16,000

The midcourse phase of an intercontinental ballistic missile begins when it leaves the atmosphere. Conventionally, this altitude is set at 100 km, where the density of the atmosphere is practically zero [19,20].

The aerodynamic drag force to which a missile (or any aircraft) is subject is given by Equation (Equation 1), where *C* is the aerodinamic drag coefficient of the missile, ρ is the density of the medium, *A* is the cross-section perpendicular to the direction of motion and *v* is the speed.
(1)$$D=\frac{1}{2}C\rho Av^{2}$$

Since the drag force is directly proportional to the air density, the drag is zero during the midcourse phase. Therefore, the missile loses very little energy during this phase. This is the key to the very long range of intercontinental ballistic missiles and the reason for using a parabolic trajectory instead of a straighter and therefore shorter trajectory. The midcourse phase is the longest phase of a missile’s flight and the phase in which the missile travels the greatest distance [12,21].

The absence of air resistance also plays a critical role in missile defense. While the missile is easy to detect in the boost phase thanks to the large amount of heat emitted by the rocket boosters, the midcourse phase is relatively ”cold”. Since the missile does not use boosters during the midcourse phase and does not emit heat when flying without drag, it is much colder than in the other phases and therefore more difficult to detect with thermal sensors. On the other hand, the space in which the missile is moving is also very cold and much colder than the missile itself. Therefore, the missile is not completely undetectable.

The last phase is the re-entry or terminal phase. It begins when the reentry vehicle re-enters the atmosphere, i.e., conventionally below 100 km altitude. Unlike the midcourse phase, drag plays a major role here because the density of the atmosphere increases significantly as the missile approaches the target on Earth. During this phase, the reentry vehicle emits a large amount of heat due to aerodynamic drag and can therefore be easily detected by thermal sensors. To our knowledge, for most missiles the duration of this phase is not publicly available, but it should be very short due to the very high terminal velocity of the missile [9,13]. According to Fletcher [12], an approximate duration of the terminal phase of intercontinental ballistic missiles is 2 min, as opposed to the midcourse phase, which is estimated to last 20 min [12].

Ballistic missiles are usually launched from land, from silos or mobile launchers on wheels or from submarines [4]. More recently, an air-launched ballistic missile, the Kh-47M2 Kinzhal, has been deployed [22].

Ballistic missiles can theoretically maneuver, i.e., change its trajectory, only during the boost phase. This is one of their major disadvantages, because it makes their trajectory extremely predictable and therefore easier to intercept. For this reason, variants of ballistic missiles have been introduced that can also maneuver during the midcourse or re-entry phase.

Multiple Independently targeted Reentry Vehicles (MIRVs) are multiple reentry vehicles installed on a single missile. After the boost phase, the tanks and boosters separate from the missile. However, with MIRVs the remainder is not a single vehicle, but a bus carrying multiple reentry vehicles, each with its own warhead. The vehicles are then released during the midcourse phase (Figure 2). The bus can easily change its trajectory thanks to small boosters, allowing the vehicles to be directed at different targets [4,17,23,24]. MIRVs present two challenges to defense systems. First, the trajectory of the missile(s) becomes much less predictable, as small corrections at very high altitudes during the midcourse phase correspond to large distances on Earth. This is a radical change because the greatest weakness of ballistic missiles is the predictability of their trajectory. In addition, the cost of missile defense increases significantly as multiple interceptors are required to intercept a single missile that can carry up to 10 re-entry vehicles [25].

Maneuverable Reentry Vehicles (MaRVs) are a variant of reentry vehicles that can maneuver during the re-entry phase by manipulating their aerodynamic lift, much like an aircraft [23,26,27,28]. As a result, the trajectory of this type of vehicle, an example of which is shown in Figure 3, is highly unpredictable and therefore very hard to intercept. In addition MaRVs could perform evasive maneuvers to avoid possible interceptors.

The trajectory shown in Figure 1 is the lowest-energy trajectory and therefore maximises range. However, a missile may follow other trajectories that correspond to a shorter maximum range but may have other advantages. In particular, a missile can fly on what is called a “depressed trajectory” [13,29]. Such a trajectory has a much lower apogee with respect to the minimum-energy one. Therefore, it becomes shorter. However, since the missile is no longer on a minimum energy trajectory, either the range or the payload must be reduced, leaving the other characteristics unchanged. The advantage of this type of trajectory is that the travel time is much shorter. However, since the range is also greatly reduced, this trajectory is most suitable for submarine-launched missiles that can approach the target and less suitable for land-based intercontinental ballistic missiles.

### 2.2. Cruise Missile

Cruise missiles differ from ballistic missiles primarily in the type of engine used. Namely, they use different types of air-breathing engines rather than rocket engines that use oxidants other than air [30]. For this reason, cruise missiles cannot fly out of the atmosphere because there is no oxygen for combustion. On the other hand, their weight and size are smaller because they are not forced to carry the oxidizer in the tank along with the fuel. Since they fly completely in the atmosphere, they are constantly exposed to air resistance and use the engines during the entire flight duration. On the other hand, the flight of a cruise missile is sustained mostly by aerodynamic lift [30]. A cruise missile can also be considered as an unmanned expendable aircraft.

While cruise missiles have a less efficient trajectory than ballistic missiles, they are much more maneuverable and their trajectory is therefore much less predictable. They can also fly at very low altitudes and are therefore much less visible to ground radars [31]. On the other hand, they typically have a shorter range, a lower payload and are slower than ballistic missiles, but consequently are also easier to hide and have a smaller radar cross-section than manned aircrafts [32]. It should be noted that, similar to ballistic missiles, the cruise missile category includes a very wide variety of weapons, from short-range tactical missiles to strategic intercontinental ones [25]. Flight speeds can also vary widely, ranging from a low 160 km h^−1^ to Mach3 [30]. Because they are subject to drag and use thrusters throughout flight, cruise missiles generally have a detectable thermal footprint. However, this is usually much less visible than that of a ballistic missile [33]. Cruise missiles can be launched from a variety of platforms, but are often launched from aircraft, a solution that mitigates the effects of their shorter range compared to ballistic missiles.

Traditionally, cruise missiles had lower speeds than ballistic missiles. More recently, however, cruise missiles capable of hypersonic speeds have been introduced [34]. These are of concern in several respects because they combine the high manoeuvrability of cruise missiles with high velocities.

### 2.3. Hypersonic Glide Vehicles

For decades, missiles were classified in the latter two categories: either ballistic or cruise. More recently, however, a new type of weapon has been introduced, Hypersonic Glide Vehicles (HGVs), often incorrectly referred to simply as “hypersonic missiles”. An HGV is a special type of reentry vehicle carried by a missile. These do not fly along a parabolic trajectory but, once released start to glide toward the target [34]. Figure 4 depicts a schematic drawing of the trajectory of an HGV. The term glide should not be misleading: since they are accelerated up to Mach 20 before being released, glide flight can last thousands of kilometers and reach intercontinental targets. HGVs fly in the atmosphere, otherwise gliding would not be possible, and can maneuver thanks to aerodynamic lift. Their main advantage over ballistic missiles is that their trajectory is unpredictable and they can fly at much lower alitutdes, so they are seen later by ground radars [26]. On the other hand, flying at hypersonic speed in the atmosphere dissipates a huge amount of heat due to drag [35]. Therefore, HGV should have a very detectable thermal print.

It should be noted that the distinctive feature of HGVs is not their hypersonic speed, as is often erroneously claimed. In fact, ballistic missiles are also capable of such speeds. Instead, their main advantage is their low-altitude flight and manoeuvrability. In this sense, HGVs are not so different from MaRVs, which, however, can manoeuvre only in the terminal phase and fly at much higher altitudes for most of the flight.

The countries that have developed HGVs call them a revolutionary achievement. However, the scientific community, while acknowledging their dangerousness, doubts that they are such a revolutionary weapon. In particular, the main criticisms concern their manoeuvrability, which is necessarily limited due to the physical constraints of flying at hypersonic speeds in the atmosphere and their alleged undetectability [26,27,34,35].

## 3. Missile Defense

A detailed description of missile defense systems is beyond the scope of this work. However, to understand the peculiarities and limitations of current sensor systems, a brief introduction is necessary.

A missile defense system consists of three main components: the sensors, the interceptors and the command and control systems. Although laser-based weapons have been proposed [1], current missile defense systems use other missiles as interceptors. Often, interceptors do not carry an explosive warhead, but rely only on kinetic energy to destroy a threat. This strategy is commonly known as ”hitting a bullet with a bullet” [36]. Of course, to intercept a threat, the sensor system must first detect it. However, using a missile as an interceptor introduces one of the most important problems in missile defense today: the interceptor needs time to reach the threat. In the case of intercontinental missiles, the defense systems and thus the interceptor launch sites, are often very far from the source of the threat. Therefore, the interceptor must travel a great distance before destroying the threat. This means that it is not enough to detect a threat. Instead, it is necessary to track it and predict its trajectory. Interceptors, like any other missile, have limited maneuverability. Therefore, we must direct them to where we suspect the threat will be at the time of interception.

Missile defense sensors can be used mainly in two ways: as early warning sensors or as tracking sensors. Early warning sensors are often satellite-based and have a wide field of view. They are used to detect a threat; however, their resolution is usually not sufficient for accurate tracking. Tracking sensors are often cued by early warning sensors and provide accurate tracking and prediction of the trajectory of the threat. In addition, interceptors are often equipped with on-board sensors that guide them during the last part of the flight. However, these types of sensors are beyond the scope of this article.

Missiles can be detected during different phases. However, long-range missiles usually also have a high velocity and their terminal phase is very short. Therefore, detection in the terminal phase usually does not allow effective interception unless the interceptor is launched near the target of the threat. In addition, debris from the destroyed threat may fall on the defended area [21]. However, terminal phase interception is successfully used against shorter range missiles [6,7]

Interception during the boost phase is relatively easy because the missiles in this phase have an appropriate thermal footprint. However, destroying the threat during this phase is often not possible because it usually occurs very far from the target country and land-based interceptors are usually not fast enough to reach the threat during this phase [37]. Reliable interception of intercontinental-range threats could be made possible by using different types of interceptors, which are difficult. Boost-phase intercepts are desirable for two reasons. First, they intercept the threat very early and thus far from their target. Second, many different types of countermeasures to fool defenses are not deployed until the midcourse phase [1,2,3,37].

Ultimately, midcourse intercept is the strategy actually used against intercontinental threats. While detection and tracking is more difficult than in the other phases, the relatively long duration makes this phase the most appropriate for the moment. This strategy is used by the U.S. Ground Based Midcourse Defense (GMD) system [37].

In the following, we provide an overview of the two kinds of sensor that are currently used for long-range missile defense.

## 4. Radar

Radar (Radio Detection and Ranging) is a kind of sensor that uses radio waves to measure the distance or speed of an object.

A radar consists of a transmitter and an antenna: a radio wave of a specific frequency is transmitted, this wave is then reflected by an obstacle and returned as an echo, which is picked up by the antenna. Since the speed of an electromagnetic wave is known, it is possible to determine the distance of the object that reflected the radio wave by measuring the time that elapses between the emission and the reception of the wave.

The spectrum of radio electromagnetic waves is quite broad; however, the most commonly used frequencies for radars range from 5 MHz to 95 GHz [38]. The frequency used by a radar determines its characteristics. Frequency ranges with similar characteristics have been grouped into bands, whose acronym (also known as band designation) is shown in Table 3.

The two most important factors affected by wave frequency are achievable range and resolution. The range is mainly affected by the atmospheric attenuation of the radio waves, which increases with frequency. Therefore, waves with lower frequencies can reach longer distances. On the other hand, resolution also increases with frequency. Therefore, to achieve high resolution, a high frequency wave is required. For these reasons, the choice of frequency and thus the band in which the radar operates, is the result of a trade-off between range and resolution.

Radars using frequencies from 3 MHz to 1 GHz (HF, VHF and UHF bands) can reach long distances, but have low resolution and precision and are not commonly used, except for the HF band, which is used for over-the-horizon radars [39,40]. The L, S, C and X bands are most commonly used in modern radars because they provide a good compromise between achievable range and resolution. In particular, the S and X bands are of great interest for missile defense. K_U_, K, K_A_, V and W can achieve high precision, but their range is lower than at lower frequencies, mainly because of atmospheric attenuation of radio waves, which is very noticeable at these frequency ranges [38].

The basic operating principle of radars is only capable of detecting objects in a certain direction. To enable detection of objects in different directions, radars often have moving parts so that the antenna can rotate around itself and have a 360° view of the surroundings. This allows it to measure both the distance of the object and the azimuth angle of rotation, which is the angle at which the radar detected the object. An important aspect of radars is the width of the transmitted radio beam: if the cone of the beam is wide, the radar can only measure the range and azimuth of the object, as shown in Figure 5a. However, 3D radars that emit narrow beams can detect multiple objects at the same azimuth and determine their elevation by measuring the orientation of the returned echo, as shown in Figure 5b. Because 3D radars often have only a single narrow beam, objects at different elevations are not detected simultaneously. Instead, the field of view is scanned at a very high frequency by a single moving beam. To work properly, the scanning frequency must be high enough with respect to the speed of the objects.

The width of the beam affects the resolution of the radar, where resolution is the minimum distance between two bodies so that they can be detected as distinct. The width of the beam is related to the so-called directivity of an antenna. This is defined as the ratio between the radiated power in a given direction and the total radiated power. Thus, to achieve a narrow beam, the directivity along the beam direction must be high. The directivity can be calculated according to Equation (Equation 2), where η is the efficiency of the antenna, *A* is the area of the antenna and λ is the wavelength.
(2)$$D=\frac{4\pi \eta A}{2\lambda }$$

Phased array radars are an alternative to radars with mechanical moving parts for 3D detection. Phased array radars have no moving parts. They consist of an electronically scanned array of antennas that transmit with different phases. By appropriately modulating the phases of the transmitted signals, they can change the direction of the resulting beam without actually rotating the radar [41]. Because there are no moving parts, phased array radars are much faster than conventional radars and can scan their field of view at high frequency. However, they have the disadvantage of a limited field of view because they typically do not rotate around themselves. Therefore, to obtain a 360° view, multiple phased array antennas are used, oriented in different directions.

### 4.1. Radar Use in Missile Defense

Radars are used in missile defense both for *detection* and *tracking* of threats.

An important parameter in selecting a radar for missile defense is resolution. As mentioned earlier, the resolution of a radar determines whether it is possible to distinguish different targets that are close together. This is particularly useful for detecting multiple targets in the midcourse or reentry phase, such as MIRVs.

The resolution of a radar is directly related to the directivity of the antenna. As shown in Equation (Equation 2), the area of the antenna must be increased to achieve the same directivity at lower frequencies. For this reason, the S and X bands are commonly used for missile defense. The main difference between the two bands is the resolution that can be achieved. Because of the frequency used and the relationship between frequency and directivity, X-band radars can more easily achieve better resolution than S-band radars. However, using radars with lower frequencies has some advantages: waves with a longer wavelength are less attenuated by obstacles such as rain and fog. Since the power of the radar echo is usually low anyway, a longer wavelength increases the signal-to-noise ratio, which increases the maximum usable range.

The major limitation of long-range missile defense radars is that they can only detect objects in the line of sight. The curvature of the Earth limits the maximum distance that can be observed with a radar. This phenomenon is commonly known as radar horizon. The maximum distance *D* of an object at height *H* that can be measured by a radar at height *h* is given by Equation (Equation 3), where *R* is the Earth’s radius and the factor *k* represents that radio waves propagate slightly outside the line of sight due to atmospheric refraction. The factor *k* is usually approximated by 4/3, assuming a radar at 700 m altitude. However, this factor is highly dependent on the altitude of the radar because the density of the atmosphere varies greatly. A suitable *k* value for radars at sea level is 2 [42].
(3)$$D=\sqrt{2kRh}+\sqrt{2krH}$$

Assuming h=10 m and k=2, a missile flying at an altitude of H=100 km would be seen at a maximum distance of 1612 km, which becomes 1253 km if the missile is at an altitude of 60 km and 2540 km if at an altitude of 250 km. While these distances may seem large, the speed of the missiles must be considered. While exact speeds during the different phases are not disclosed, we know, for example, that a Minuteman III, an intercontinental ballistic missile, can travel 12,000 km in 30 min [43]. As can be seen, the maximum measurable distance also depends on the altitude of the object being detected. This complicates the detection and tracking of HGVs and cruise missiles, since they fly at low altitudes, unlike ballistic missiles.

There are ways to improve the maximum distance a radar can reach. One example is to use a special type of radar, i.e., over-the-horizon radar. There are different types of over-the-horizon radars, but by far the most popular uses sky wave propagation. Over-the-horizon radars exploit the fact that electromagnetic waves are reflected from the ionosphere back to the Earth’s surface; the amount of reflection depends on the frequency. Therefore, they use waves in the HF band, where the reflection is relevant, but an acceptable resolution can still be achieved.

With over-the-horizon radars, distances far beyond the line of sight can be achieved [39,40]. While overcoming line-of-sight limitations is a major advantage, over-the-horizon radars also have disadvantages. First, antennas of considerable size are required to produce a narrow beam at HF band frequencies: as shown in Equation (Equation 2), lower frequencies require a larger antenna area to achieve high directivity. In addition, the echo generated by these radars usually has a very low intensity. Finally, reflection from the ionosphere, even if it allows the radar to see beyond the visual horizon, is a source of noise. In addition, the atmosphere must be characterized accurately and in a timely manner to obtain good results. All these phenomena contribute to the fact that over-the-horizon radars are inaccurate and have very low resolution, barely comparable to that of conventional radars in the bands between L and X. They are, therefore, hardly used for tracking and intercepting a missile, an application that requires high accuracy and resolution. However, they can be used as early warning sensors.

Another way to overcome line-of-sight limitations is to place the radars on mobile platforms so that they can be transported and placed closer to the launch site. One option for mobile radar placement is ship-mounted radar: this solution makes radars much more mobile and flexible, allowing them to get closer to potential threat sources. Example of this techniques are the U.S. Sea-Based X-band Radar [44] or the AN-SPY1 radar of the U.S. Aegis Combat System [45].

It is worth noting there are also *space-based* radars, i.e., radars mounted on satellites. These can be used for civil purposes, such as radars for meteorological purposes, as well as for military purposes. As for the latter, they do not currently appear to be used for missile defense, but rather for reconnaissance or intelligence. One system currently under development is GMTI (Ground Moving Target Indicator), a space-based radar system for tracking objects on the Earth’s surface [46]. There is not much information about GMTI and while it does not appear to be directly aimed at missile defense, it could still be used to detect and track large mobile launch platforms [31]. The potential of space-based radars has already been widely recognized:

“*As an overarching finding, the task force believes that the Space-Based Radar has the potential for substantial contributions to ballistic missile defense, providing capabilities and access that are difficult to achieve with surface-based sensors.*”[47]

Of particular importance in the context of space-based radars are Synthetic Aperture Radars (SARs). They are mounted on moving platforms, usually satellites or aircraft and can simulate a large antenna by moving the platform [48]. As mentioned earlier, the directivity of an antenna also depends on its size. Thus, the ability to simulate a large antenna allows for high resolution. A radar of the SAR type makes it possible to obtain a detailed radar image of the area by moving for a period of time *T* and storing the characteristics of the detected echoes. The results are obtained as if an antenna of size T·v were used, where *v* is the speed of the platform. This type of radar is commonly used for air defense, but it does not appear to be used for long-range missile defense at this time, although its use has been proposed and studied [47].

### 4.2. Radars in Missile Defense Systems

Radars are used in a variety of missile defense systems. In this section, we briefly present some examples that we consider relevant in the context of radar sensing. It should be noted that only a fraction of these systems were used in real warfare. Moreover, some were used only a few times. Therefore, an objective evaluation of their effectiveness is not always possible.

The MIM-104 Patriot system is the U.S. main air and short-range missile defense system [5]. It uses an AN/MPQ-53 radar which has recently been upgraded to an AN/MPQ-65. Both are phased array radars that use frequencies in the C-band and are used to detect and track threats up to 100 km away.

The Patriot system has been used in a number of real-world defense operations and has been sold to several other countries outside the United States. The first deployment was in 1991 during Operation Desert Storm [7,49]. More recent deployments were during Operation Iraqi Freedom in 2003 [49] or in Saudi Arabia to defend its own oil sites [50]. However, the effectiveness of the Patriot system in these operations is debatable [7,49,50,51].

In addition to the U.S., other countries have developed similar radars. One example is the Chinese H-200 radar, which has target detection, tracking and missile guidance capabilities with a range comparable to the AN /MPQ-53.

The THAAD (Terminal High Altitude Area Defense) is a U.S. mobile defense system used to intercept ballistic missiles and other airborne threats [6]. It uses an AN/TPY-2 radar, which is a phased array radar that uses frequencies in the X-band. The use of X-band waves provides higher resolution compared to lower frequencies, allowing targets to be distinguished from decoys. While it is theoretically possible to achieve the same resolution at lower frequencies, this would require larger antennas, which would be a disadvantage since THAAD is a mobile system. The AN/TPY-2 can be used in two modes: terminal mode or forward-based mode. In terminal mode, it can be used to track and intercept missiles and has a range of 1000 km, while in forward-based mode it can be used in conjunction with other defense systems and can reach a range of 2000 km. It is effective against SRBMs, MRBMs and IRBMs, but its use against long-range missiles is severely limited by its sensing capabilities [52].

The THAAD systems has also been deployed outside the United States; for example in South Korea [53] or in the United Arab Emirates [54], where it was used for the first (and only) time in combat [55].

A Chinese alternative to the THAAD is the HQ-19 system, which is capable of intercepting ballistic missiles outside the atmosphere. It is an updated version of the HQ-9 system, which can intercept only short-range ballistic missile at a range of up to 500 km [56]. In contrast, the estimated range of the HQ-19 is between 1000 km and 3000 km [57]. In addition, it is also believed to have anti-satellite capabilities [56]. This means that it could be used to destroy satellites in low-Earth orbit. The Russian Federation has also developed an alternative to the THAAD system, the S-400 [58].

The U.S. AEGIS Ballistic Missile Defense system is part of the U.S. naval AEGIS combat system. It aims to intercept ballistic missiles during mid-course and reentry phases and can also be used against cruise missiles and aircrafts [59]. The fundamental sensor of the system is the SPY-1 radar, a phased array radar operating in the S-band. It uses four planar antennas to provide a 360° view and overcome the limited field of view of phased array radars. This radar has a maximum range of 370 km [60]. An updated version of the SPY-1 is the SPY-6 radar, which actually consists of two different radars integrated into a unique system. The first radar operates in the S-band and is used for missile and aircraft detection and tracking. The second radar operates in the X-band and is used for terminal illumination of targets, a technique in which the radar emits radio waves that are detected by sensors on the interceptor [61] and used for guidance. Compared to the SPY-1, the SPY-6 offers greater sensitivity and range.

The Ground-based Midcourse Defense (GMD) is the U.S. defense system against intercontinental ballistic missiles. It does not rely on a single sensor but integrates data from multiple sensors [1]. As for radars, it can use the Cobra Dane radar [41] and also integrates radar data coming from other systems, e.g., SPY-1 radar from the AEGIS system. Another radar of interest is the Sea-Based X-Band Radar-1 (SBX-1): This is a floating radar platform that operates in the X-band. It has high resolution, which makes it useful for distinguishing between reentry vehicles and decoys, but it also has a limited field of view that makes it unsuitable for tracking targets [44].

Also of interest is Russia’s A-135 missile defense system, which uses a long-range Don-2N radar and was designed to defend the area surrounding Moscow [62]. However, very little information about this system is publicly available.

## 5. Space-Based Infrared Imaging Sensors

Infrared imaging sensors are the most common kind of space-based sensor used for missile defense. These, also known as thermal imaging sensors, are imaging sensors sensible to the infrared spectrum, much like an rgb camera is sensible to visible light. They are not to be confused with Passive Infrared Sensors (PIRs), which are much simpler, do not produce an image and are not particularly useful in the field of missile defense. Infrared imaging sensors belong to two different categories: either photon detectors or thermal detectors. The former have high sensibility and a fast response; however, they need to be cooled and are, therefore, big, heavy and costly. The latter are less sensible and have a slower response; however, they do not need to be cooled [63]. Both kinds of sensors can be used on satellites for missile defense.

Even though the term is not completely accurate, for the sake of simplicity, in this work we will refer to infrared imaging sensors simply as infrared sensors.

The working of an infrared sensor can be easily explained using Plank’s Law describing the spectral radiance of a black body at thermal equilibrium:(4)$$B\left(\nu ,T\right)=\frac{2h\nu^{3}}{c^{2}}\frac{1}{e^{\frac{h\nu }{K_{B}T}}-1}$$
where ν is the frequency of the spectral radiance, *T* the temperature, $K_{B}$ the Boltzmann constant, *h* Plank’s constant and *c* the speed of light. From Plank’s law we can derive how increasing the temperature of the body shifts the peak frequency higher. This is particularly evident if we consider Wien’s Displacement Law, which can be derived from Plank’s Law:(5)$$\lambda_{peak}=\frac{b}{T}$$
where *b* is Wien’s displacement constant.

Wien’s and Plank’s law is especially important in the field of missile defense for two reasons. First of all, it clarifies why infrared sensors are necessary, as opposed to normal rgb cameras. Indeed, the peak frequency of radiant energy lies in the spectrum of visible lights only for very high temperatures, more or less above 4000 K. Therefore, visible light cameras are not suitable for detecting objects that are not illuminated. On the other hand, for more common temperatures the peak frequency lies in the infrared spectrum.

Second and most important, Wien’s Law allows the selection of the best infrared band depending on the expected temperature of the object to be detected. As explained earlier, the temperature of a missile varies significantly during its flight and between categories. In addition, infrared sensors typically do not cover the entire spectrum (although multiband sensors do exist). Therefore, different sensors are suited to the detection during different phases and of different categories of missiles and the choice of band is critical. The bands that make up the infrared spectrum are listed with their abbreviations and corresponding frequency range in Table 4. Ballistic missiles can be easily detected in the boost phase thanks to the heat generated by the rocket engines. In the midcourse phase, however, the missile flies without propulsion in an environment without air resistance. Therefore, it is more difficult to detect because it does not give off much heat and is much colder. Finally, the terminal phase is similar to the boost phase. Although the reentry vehicle does not use rocket engines in this phase, it still gets very hot at very high speeds because of the drag. It should be noted, however, that infrared sensors are of little use during this phase. The reason is that in the terminal phase, the missile is usually within range of radars, which are required for the accurate tracking by the interceptor anyway. In addition, detection of an intercontinental ballistic missile only in the final phase usually does not permit effective defense because there is little time for intervention.

Cruise missiles are fundamentally different from ballistic missiles. Namely, they are continuously propelled by jet engines. Therefore, they have an infrared signature that can be easily detected with infrared sensors [11,32,64]. The only obstacle could be cloud cover when the rockets are flying at low altitudes. In addition, the recently introduced hypersonic cruise missiles dissipate a large amount of heat due to atmospheric drag [27].

The lift phase of an HGV is essentially the same as ballistic missiles, so the same considerations apply. During the glide phase, however, the vehicle is easily seen due to the enormous amount of heat dissipated by flying at supersonic speeds in the atmosphere [27,34,35].

In summary, the only phase with very peculiar properties is the phase in the middle of the course of a ballistic missile. In this context, the LWIR band is more suitable for detecting relatively cold objects, in contrast to the SWIR and MWIR bands, which are more suitable for the ”hotter” phases [14,17]

The choice of frequency band is also critical for optimizing the signal-to-interference ratio. Schweitzer et al. [14] analyzed the frequencies at which the absorption by the atmosphere of infrared waves radiated from the Earth’s surface is maximum [14]. Contrast is critical for accurate detection: if the missile has good contrast with the background, it is much easier to detect. For infrared sensors on satellites, the background is usually represented by the Earth’s surface. Therefore, it is important to isolate the radiation from the missile from the radiation from the Earth’s surface. The Earth’s atmosphere absorbs the radiation, so the best strategy is to choose an infrared band whose absorption by the Earth’s atmosphere is maximum. In this way, the radiation from the Earth is absorbed by the atmosphere, while the radiation from the missile, which is in the upper layers of the atmosphere where the absorption is less, is clearly visible. The bands centered at 2.7 μm, at 4.5 μm and at 6.3 μm are optimal in this sense. In addition, the authors also analyzed the effect of scattering from the solar radiation cloud cover: below 2.5 μm, scattering from the *cumuli* produces a strong signal that can interfere with detection [14].

During the midcourse phase, one strategy to maximize the signal-to-clutter ratio is to view the missile from a vantage point that includes space as a background rather than the Earth’s surface [14,17]. However, this strategy results in a reduction in the useful field of view of a satellite, requiring more satellites to provide the same coverage.

### 5.1. Space-Based Infrared Sensor Systems for Missile Defense

Since the 1950s, the very limited advances provided by the radard-based Ballistic Missile Early Warning System brought the USA to investigate satellite-based solutions [65]. The first project to address this issue was the Missile Defense Alarm System (MIDAS). It should have been composed of 12 satellites in polar orbit that should have provided continuous coverage of the Soviet Union. Based on infrared sensors, its goal was to detect missile launches and atomic detonations [65]. However, the project did not succeed as expected, mainly due to the sensor technology available at the time [66]. The first operative infrared sensor satellite network for missile defense was the Defense Support Program (DSP). Based on infrared sensors on the MWIR band, it was used to detect missiles during the boost phase [4]. Wide field of view sensors were used, which enabled a complete Earth coverage with only three satellites; however, they did not allow a precise tracking of the missiles and were used only for early detection [67].

The successor of the DSP project is the Space-Based Infrared System (SBIRS), which is composed of satellites on geostationary orbit (SBIRS-GEO) or high elliptical orbit (SBIRS-HEO), equipped with infrared sensors sensible at the SWIR and MWIR bands [68]. From 2006 to 2021 nine satellites were launched: four SBIRS-HEO and five SBIRS-GEO. The main improvements in the SBIRS network, with respect to DSP, are the sensors used. Each satellite, indeed, is equipped with two infrared sensors: a scanning sensor and a step-starter [69]. The former has a wide field of view and is used for the early detection of missile threats; however, it cannot be used for precise tracking. For this task, the step-starter sensor is used, which has a narrower field of view and is cued by the scanner. This combination should allow SBIRS to go beyond just early warning. It has to be noted, however, that the sensors used are mainly suited to the detection of high temperature objects and, therefore, have limited utility during the midcourse phase [67]. Initially, SBIRS should have included a third group of satellites in low Earth orbit (SBIRS-LOW). However, the project was transferred from the U.S. Space Force to the Ballistic Missile Defense Organization (now Missile Defense Agency) and renamed Space Tracking and Surveillance System (STSS) [70]. Differently from SBIRS, STSS should allow the precise tracking of missiles during each phase of flight. However, it was mainly conceived as a research program rather than operational and, therefore, only two satellites have been launched. The program which should have succeeded STSS, the Precision Tracking Space System, has instead been cancelled. Very limited information is available on more recent programs, such as the HBTSS (Hypersonic and Ballistic Tracking Space Sensor), whose first satellite is planned to launch in March 2023. It is designed to allow the precise tracking of ballistic and hypersonic glide vehicles and should also exploit information from other sensor networks such ash SBIRS [71]. The same considerations holds also for the Next-Generation Overhead Persistent Infrared (Next-Gen OPIR) system, whose launch is planned for 2025. It will be composed of satellites on geostationary or highly elliptical orbit [72] and will be equipped with sensors with higher resolutions than its predecessors [73].

It is very likely that other countries besides the U.S. have space-based infrared sensor systems for missile defense, especially Russia and China. However, no public and reliable information about these systems is available.

### 5.2. Infrared Sensors and Countermeasures

A very important research question is whether infrared sensors are able to distinguish between a reentry vehicle delivering a warhead and countermeasures, i.e., devices used to deceive defense systems. Most types of countermeasures are used during the midcourse phase, the most common being Mylar balloons [1,2]. The concept of midcourse countermeasures is to saturate the defense systems with too many interceptor targets that should be very difficult to distinguish from the true reentry vehicle [3], cheap enough to produce (at least cheaper than an interceptor) and light enough not to reduce the missile payload too much. Balloons deployed during the midcourse phase meet all of these characteristics. Since there is no air resistance at this stage, the trajectory and speed of an object are not affected by its shape. Therefore, it is not possible to distinguish the reentry vehicle from a balloon based on its trajectory. In addition, balloons and reentry vehicles are too small to be distinguished by their shape using current missile defense sensing technology [2]. However, it is possible to do so based on their infrared signature [74]; indeed, this appears to be the strategy used in testing the GMD defense system in the United States [1,2]. However, it should be noted that the infrared signature of a balloon can be manipulated by heating or cooling [3,6]. In addition, a missile could not only deploy an empty balloon, but also hide the warhead inside a balloon, making its detection much more difficult [24]. The effectiveness of balloons is also due to the fact that they are very cheap and light. Therefore, a single missile can deploy a number of them.

Mylar balloons do not work in the terminal phase because they would be easily distinguishable from the real warhead due to air resistance. At this stage, the most common strategy is to use MIRV or MaRV. The former provide a defense system with multiple targets, some of which could even be false warheads. The latter make it much more difficult to predict the trajectory of the vehicle and thus intercept it [3].

## 6. Conclusions

While there are effective systems against short- and intermediate-range missiles, defense against long-range threats is still very much an open issue. This is partly due to interceptors and partly due to sensor technology. Despite recent advances in weapon systems, ballistic missiles are still the most important threat. Their flight can be divided into phases with very different characteristics that require different defense strategies. While the detection during the boost phase is highly advantageous due to the large amount of heat generated and the limited availability of countermeasures, interception during the boost phase is still severely limited by interceptor capabilities.

Detection and tracking during the terminal phase is also very effective because the missile is within range of radar systems. However, the very short duration of this phase makes intercepting fast-flying threats very difficult. There is also a high risk that the debris from the intercepted missile will cause damage to the defended area.

The midcourse phase presents the greatest challenges to sensor systems because ballistic missiles emit much less heat and because sensors are easily deceived by countermeasures. However, this is the longest phase, so intercepting intercontinental threats, which usually fly at hypersonic speeds, is much more feasible. For this reason, research activities on defense against intercontinental missiles have focused mainly on this phase. To our knowledge, midcourse interception is used in the U.S. GMD system, which is the only deployed system designed to intercept intercontinental threats.

Other types of threats, such as cruise missiles or HGVs, have different phases of flight but have in common the large amount of heat generated. For this reason, detection and tracking techniques for boost and terminal phases of ballistic missiles are usually applicable. However, since these weapons typically fly at much lower altitudes, it is possible that infrared sensors will need to be mounted on satellites in lower orbits. This statement is supported by the fact that the U.S. are deploying the HBTSS to defend against hypersonic gliding vehicles. Therefore, it is likely that other existing satellite networks are not suitable for this task.

Two main types of sensors are still used in missile defense systems: radars and space-based infrared imaging sensors. These are not alternatives, but complement each other. We have summarized the characteristics and differences of radars and infrared sensors in Table 5.

Radars are still the best choice for tracking. Even though tracking seems possible with the latest space-based infrared systems, the precision and accuracy of radars is still difficult to achieve. For this reason, they are still the best choice for interceptor guidance, a step that is obviously necessary for a missile defense system. There are many radar types to choose from. They range from relatively small, portable radars such as those in the Patriot and THAAD systems, which are best suited for short- and medium-range missile defense, to huge long-range radars such as the Cobra Dane. In addition, radars can be ground-based but also installed on ships, such as those of the AEGIS Combat System or on aircraft, although the latter are mainly used for airspace. There are also space-based radars, but to our knowledge they are used for recoinnaisance, rather than missile defense.

On the other hand, radars still suffer from an important limitation: the radar horizon. That is, they cannot see beyond the horizon. This severely limits their use as an early warning system against intercontinental threats. Moreover, for the same reason, they are not suitable for boost-phase defense, which, although not yet implemented, would be the best defense strategy. While sea-based radars and the possibility of deploying ground radars on the territory of allied countries can reduce the impact of this limitation, space-based infrared sensors are still the best solution to overcome the radar horizon problem.

With a sufficient number of satellites, an infrared satellite network can easily cover the entire surface of the Earth. For this reason, it is very well suited for early warning tasks, as it can detect a missile shortly after launch. On the other hand, current sensor technology does not yet allow the precise tracking required for interception. Therefore, they are more of a complement to radar systems than an alternative. A common strategy against intercontinental threats is to use cues from infrared systems in conjunction with radar systems to intercept a missile during the midcourse phase. The ability to detect missiles shortly after launch may seem minor with respect to the reduced precision, but it is particularly relevant with more recent threats such as hypersonic gliding vehicles. Their hypersonic speed, combined with their ability to fly at low altitudes, makes the radar horizon problem even more relevant. Indeed, radars might not detect these types of threats until it is too late to intercept them.

Space-based infrared sensor systems also have drawbacks. First of all, their cost is still very high, especially if full coverage of the Earth is required. However, the cost of launching satellites is expected to decrease in the future [73]. In addition, they could be very vulnerable to anti-satellite weapons [75], because a satellite has a very predictable trajectory, very limited (if any) maneuverability and no type of defense.

As shown by various sources, missile defense systems still have many limitations and cannot be considered reliable against long-range threats. Therefore, we believe that academic research in this area is really necessary and can foster progress in a very important field nowadays. Moreover, technical information on this topic is often fragmentary and reliable sources are scarce. With this overview, we aim to promote research activity in this field and provide clear and reliable information for academics and decision makers.

## Figures and Tables

**Figure 1 sensors-22-09871-f001:**
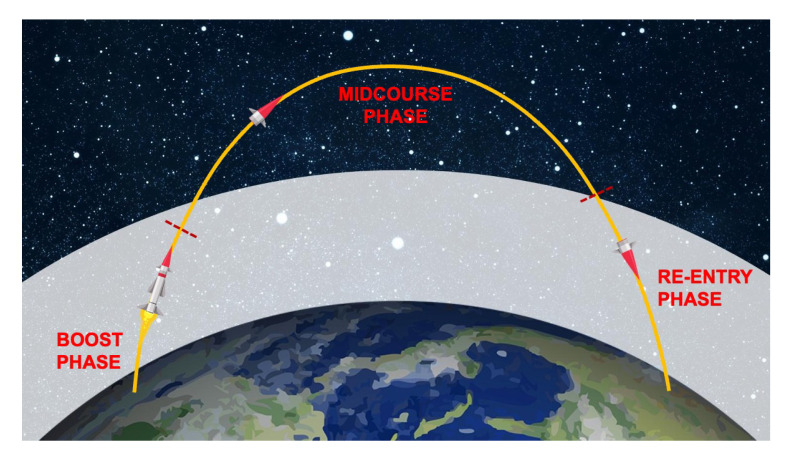
The trajectory and phases of the flight of a ballistic missile. Most of the time is spent outside the atmosphere (represented by the white area) and engines are used only during the boost phase. The red dashed lines represents the boundaries of the various phases.

**Figure 2 sensors-22-09871-f002:**
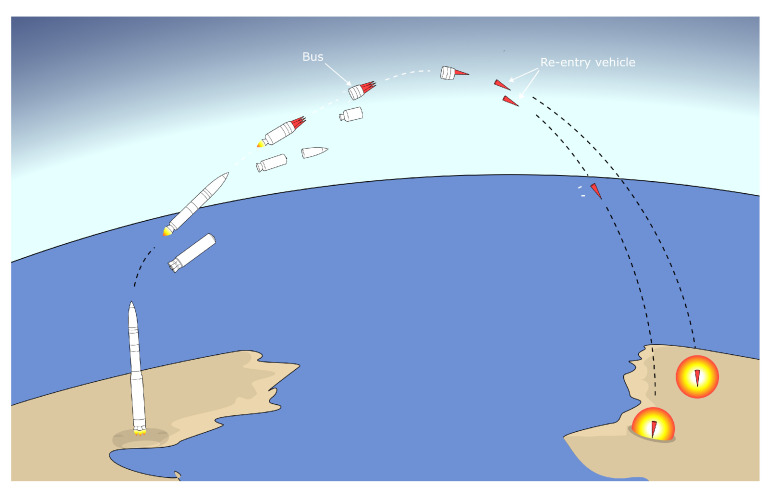
A schematic drawing of the operation of a MIRV.

**Figure 3 sensors-22-09871-f003:**
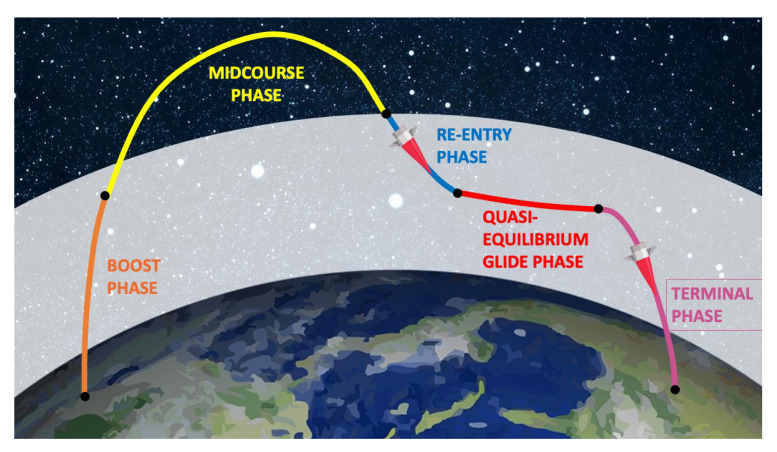
The trajectory of a Maneuverable Reentry Vehicle (MaRV).

**Figure 4 sensors-22-09871-f004:**
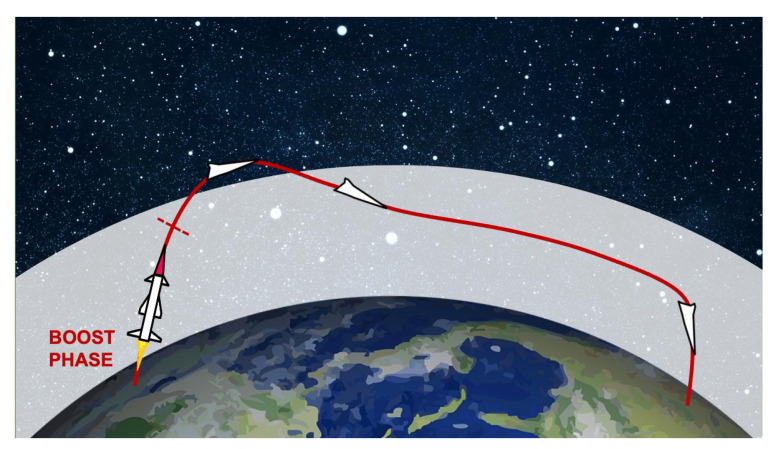
The trajectory of a Hypersonic Glide Vehicle (HGV). The red dashed line represents the boundary of the boost phase. The white area represents the athmosphere.

**Figure 5 sensors-22-09871-f005:**
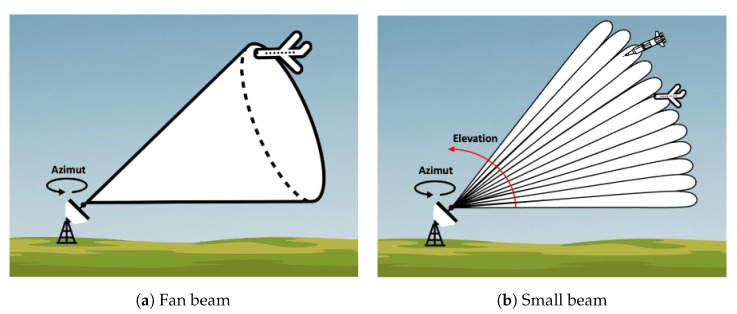
Schematic representation of two radars, one with a large beam width, also known as fan beam (**a**), the other with small beam width (**b**) and thus better directivity. The second radar can distinguish multiple objects in its field of view.

**Table 1 sensors-22-09871-t001:** Designations of ballistic missile according to the maximum range [9,10].

Name	Range (km)
Short-Range Ballistic Missile (SRBM)	<1000
Medium-Range Ballistic Missile (MRBM)	1000 < x < 3000
Intermediate-Range Ballistic Missile (IRBM)	3000 < x < 5500
Intercontinental Ballistic Missile (ICBM)	>5500

**Table 3 sensors-22-09871-t003:** IEEE band Designation and Nominal Frequency Range [38].

Band Designation	Nominal Frequency Range
HF	3–30 MHz
VHF	30–300 MHz
UHF	300 MHz–1 GHz
L	1–2 GHz
S	2–4 GHz
C	4–8 GHz
X	8–12 GHz
K_U_	12–18 GHz
K	18–27 GHz
K_A_	27–40 GHz
V	40–75 GHz
W	75–110 GHz

**Table 4 sensors-22-09871-t004:** The infrared spectrum bands, their abbreviations and corresponding wavelengths.

Name	Abbr.	Wave Length Range (μm)
Near Infrared	NIR	0.75–1.4
Short Wavelength Infrared	SWIR	1.4–3
Mid Wavelength Infrared	MWIR	3–8
Long Wavelength Infrared	LWIR	8–15
Far Infrared	FIR	15–1000

**Table 5 sensors-22-09871-t005:** Comparison of the two main types of sensor for long range missile defense.

	Infrared	Radar
Detection Phase	Boost and midcourse	Midcourse ^1^ and terminal
Early warning	Yes	No
Tracking	Yes	Yes
Interceptor guidance	No	Yes
Location	Space	Space ^2^, ground, sea, air
Coverage	Earth surface ^3^	Limited by radar horizon
Cost	Hightest	High

^1^ If closer than radar horizon. ^2^ Used for reconnaissance rather than missile defense. ^3^ With an adequate number
of satellites.

## Data Availability

Not applicable.

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
