# Peer review of "An Overview of Sensors for Long Range Missile Defense"

_sensors, 2022, doi:10.3390/s22249871_

Round 1

Reviewer 1 Report

In the manuscript entitled “An overview of sensors for long range missile defense” the authors provide an overview of the two most common types of sensors: radars and infrared satellites. The study is relevant to the journal's audience and improves our understanding of the subject. Thus, I believe that, if the manuscript is improved, it will be able to meet the requirements for publication.

1)      fix ")" in keywords

2)       Authors can standardize the way of citing "infrared satellites"

3)       The authors put a subsection "Examples of Radars" but did not put a subsection on "Examples of Space-based infrared image sensors".

4)       The authors put a subsection “History of IR-based satellites for missile defense”, but there is no subsection on the "History of Radars”.

5)       Correct the sentence: “The THAAD system has proven effective against SRBM, MRBM, and IRBM, however its use against long range treats is severely limited by its sensing capabilities”

6)      Authors should explain some details such as the bands L, S, C, X, etc.

7)      The authors have written the conclusion in a way that is very superficial and that looks a lot like the abstract. It would be important to rewrite the conclusion in more depth about the two sensors.

8)      Lots of old references. Authors can address the latest in the literature

Author Response

  • "In the manuscript entitled “An overview of sensors for long range missile defense” the authors provide an overview of the two most common types of sensors: radars and infrared satellites. The study is relevant to the journal's audience and improves our understanding of the subject. Thus, I believe that, if the manuscript is improved, it will be able to meet the requirements for publication."

We thank the reviewer for taking the time to provide a valuable and comprehensive review, which we believe has greatly helped us improve the paper.

  • "fix ")" in keywords"

  • "Authors can standardize the way of citing "infrared satellites"

Thanks, we fixed these issues.

  • "The authors put a subsection "Examples of Radars" but did not put a subsection on "Examples of Space-based infrared image sensors".
  • "The authors put a subsection “History of IR-based satellites for missile defense”, but there is no subsection on the "History of Radars”.

In fact, there is a discrepancy in the naming of the sections. In the revised version we have corrected this problem. The two sections, although originally named differently, serve the same purpose. It is true that there is a slight difference in how the two sections are written: The section on infrared is written as a history of systems, rather than as a description of the current systems. This is because of how the two different types of sensor systems have evolved. Newer radar systems have not replaced the previous ones; instead, they have been an addition to a layered system, in which each layer is used to defend against threats at a given distance.

In contrast, infrared-based systems have evolved mostly replacing the previous versions. The exceptions are reported. This is probably because infrared systems are primarily used only against long-range and intercontinental threats, a much more limited variety of threats.

  • " Correct the sentence: “The THAAD system has proven effective against SRBM, MRBM, and IRBM, however its use against long range treats is severely limited by its sensing capabilities”

Thanks, we fixed the sentence.

  • " Authors should explain some details such as the bands L, S, C, X, etc."

While there is no relevant difference in the operation of radars at different frequencies, there is indeed a difference in the outcome. Therefore, we have added details on how accuracy and range varies between frequency bands and when to choose one band over another.

  • "The authors have written the conclusion in a way that is very superficial and that looks a lot like the abstract. It would be important to rewrite the conclusion in more depth about the two sensors."

We agree that the conclusions were a bit too short and did not add much to the paper. For this reason we expanded them. We added a summary about the different properties of the sensors, about when to use one sensor versus another, and how they complement each other. Also, as suggested by Reviewer3, we added a table to summarise these observations.

  • " Lots of old references. Authors can address the latest in the literature"

Some references are indeed very old indeed. This is partly due to the lack of academic studies on the subject. However, we understand the need for more recent references and have therefore added them. We believe that some of the older references are still relevant. Therefore, rather than replacing them, we have added new ones without removing the older ones.

Reviewer 2 Report

The paper is a review that describes the type of missiles from low to very high range and methods for their detection. Different type of sensors are given which are useful for different type of missiles. The paper does not present scientific results or technological suggestions for improvement the localization capability. However the paper is well written and clear. I suggest the paper to be published after minor corrections. Here some questions/suggestions:  

Line 81 looks like a repetition of what I have read earlier.

Table 2: Even if I am not in the field I understand more or less the units but it is better to specify [km] and [s] somewhere. 

Line 85: why thr authors say a parabolic trajectory looks an illogical choice? Maybe compared with a straight line or, better,  curved and parallel to the ground  (providing shorter distance). Better to explain this phrase here with motivation given in line 109, and report again at line 109 if needed.  

Line 155: Even I understand there is a variation in the minimum energy trajectory, I do not understand the sentence "there are other possibilities". Possibilities of what? there are infinite trajectories of not minimum energy. Can the author explain better the meaning of "a much lower apogee"?

Line 170: fuel. Since (I think a period is mission

Line 171 "unlike ballistic missiles 171 that use boosters only during the boost phase" : this concept was already clear in the previous section, no need to repeat in the cruise missiles section.

Line 197  "fig. 4 depicts" -- > upper case F

Line 232 "at the time of 232 interception.." a period is missing

Table 3  Please check the frequency limits , if I am not wrong the UHF should be in the GHz

Line 274 "Radars using the lower part of the spectrum" better saying the lower frequencies

Line 288. I don t have any background on radar detectors. Can the author  describe if the 3d detection occurs in real time or if a  narrow beam must scan at different elevations in order to resolve multiple objecys as in fig 5b?

Line 312 "such as MIRVs The" period is missing

Line 345: As fas as I know LF or VLF can be reflected by the ionosfere as well. 

Line 522 The programm --> program

Line 534 "i.e., ," double comma 

Thank you for this interesting and - very important for present days - review. You refer often to the US systems for sensors [e.g. lines 360, 387, 403, 415, 493]. To your knolwdge, there are other ground or space based system used by other counties (e.g. Russia, China) ? Has Europe some proper sensors/tools or everything is in common with US through NATO?

Author Response

  • "The paper is a review that describes the type of missiles from low to very high range and methods for their detection. Different type of sensors are given which are useful for different type of missiles. The paper does not present scientific results or technological suggestions for improvement the localization capability. However the paper is well written and clear. I suggest the paper to be published after minor corrections."

We sincerely thank the reviewer for her/his appreciation and for the valuable suggestions and the detailed review.

  • "Line 81 looks like a repetition of what I have read earlier."

Thanks, we removed the sentence.

  • "Table 2: Even if I am not in the field I understand more or less the units but it is better to specify [km] and [s] somewhere. "

Measurement units are definitely needed. Therefore, we added them.

  • "Line 85: why thr authors say a parabolic trajectory looks an illogical choice? Maybe compared with a straight line or, better,  curved and parallel to the ground  (providing shorter distance). Better to explain this phrase here with motivation given in line 109, and report again at line 109 if needed.  "

Our sentence was definitely unclear. We meant that a parabolic trajectory is not the shortest (in terms of distance). We followed the reviewer's suggestion and rewrote the sentence more clearly and with more motivations.

  • "Line 170: fuel. Since (I think a period is mission"
  • "Line 171 "unlike ballistic missiles 171 that use boosters only during the boost phase" : this concept was already clear in the previous section, no need to repeat in the cruise missiles section."
  • "Line 197  "fig. 4 depicts" -- > upper case F""
  • "Line 232 "at the time of 232 interception.." a period is missing"
  • "Line 274 "Radars using the lower part of the spectrum" better saying the lower frequencies"
  • "Line 312 "such as MIRVs The" period is missing"
  • "Line 522 The programm --> program"
  • "Line 534 "i.e., ," double comma"

Thanks, we fixed these sentences.

  • "Table 3  Please check the frequency limits , if I am not wrong the UHF should be in the GHz"

Thanks, there was an error in the frequency limits of the UHF band.

  • "Line 288. I don t have any background on radar detectors. Can the author  describe if the 3d detection occurs in real time or if a  narrow beam must scan at different elevations in order to resolve multiple objecys as in fig 5b?"

With 3D radars, a narrow beam is moved (either physically or electronically ) very quickly to scan the entire field of view. Therefore, a single beam is used to scan at different elevations to resolve multiple objects. We have made this point clearer in the text.

  • "Line 345: As fas as I know LF or VLF can be reflected by the ionosfere as well." 

It is true that LF and VLF waves are also reflected from the ionosphere. However, the resolution that can be achieved at such low frequencies is generally not suitable for missile defense. We have clarified this point in the text and explained why only HF is used for missile defense, even if other waves are reflected.

  • " You refer often to the US systems for sensors [e.g. lines 360, 387, 403, 415, 493]. To your knolwdge, there are other ground or space based system used by other counties (e.g. Russia, China) ? Has Europe some proper sensors/tools or everything is in common with US through NATO?"

We refer mainly to U.S. systems, since these are the only ones for which detailed information is available. As for space-based defense systems, to our knowledge, the U.S. system is the only one currently deployed. While it is likely that Russia and China are at least researching such systems, no information is available to the public. We understand that it is important to provide information on other countries as well. For this reason, in the revised version, we have added information on Russian and Chinese systems and how they compare to U.S. systems, even though detailed data are not available.

It should be noted that U.S. systems are also deployed outside the United States. Partly because they are deployed in allied territories (e.g., North Korea), partly because they are sold to other countries (e.g., the Arab Emirates). Thanks to these two factors, Europe has some missile defense capability. In the revised version, we added some information about U.S. systems deployed abroad.

Reviewer 3 Report

The topic is interesting. This paper is well-organized and well-written.

I have some minor concerns for the authors’ consideration of a revision.

1. A table is needed to summary the peculiarities, drawbacks and most important the perspectives of the discussed two sensors.

2. It is recommended to provide some real cases of missile defense in war so as to support argues and perspectives in this paper.

Author Response

  • "The topic is interesting. This paper is well-organized and well-written."

We would like to thank the reviewer for taking the time and effort necessary to review and for her/his appreciation.

  • "A table is needed to summary the peculiarities, drawbacks and most important the perspectives of the discussed two sensors."

We agree that a table could be helpful to better understand the paper and to capture the specifics of the different sensors. For this reason, we have added Table5 where this information is presented.

  • "It is recommended to provide some real cases of missile defense in war so as to support argues and perspectives in this paper."

Fortunately, the vast majority of the defense systems presented have never been used in a real battle. In addition, some systems have probably been used, but no information is available about them. This is especially true for the Chinese and Russian systems, which we have included in the revised version as suggested by Reviewer2. On the other hand, we understand that real cases can be very important. For this reason, we have added a brief description and references to publicly available reports of real-world wartime engagements for the THAAD and Patriot systems to elaborate on the topic. To our knowledge, these are the only ones used in combat.

Round 2

Reviewer 1 Report

The authors answered all questions and queries and improved the manuscript accordingly. In my opinion, the present manuscript may be published in Sensors.